**Data Availability Statement:** The authors have shared summary data tables as allowed by the ethical clearance procedures of the CDC. However,

# A qualitative study of behavioral and social drivers of COVID-19 vaccine confidence and uptake among unvaccinated Americans in the US April-May 2021

Neetu Abad[1]☯*, Seth D. Messinger[2]☯, Qian Huang[3]☯, Megan A. Hendrich[2]☯, Nataly Johanson[2]‡, Helen Fisun[2]‡, Zachary Lewis[2]‡, Elisabeth Wilhelm[1]‡, Brittney Baack[1]‡, Kimberly E. Bonner[1]‡, Rosemarie Kobau[1]‡, Noel T. Brewer[3,4]☯

**1** US Centers for Disease Control and Prevention, Atlanta, GA, United States of America, **2** Ipsos US Public Affairs, Washington, DC, United States of America, **3** Gillings School of Global Public Health, University of North Carolina, Chapel Hill, NC, United States of America, **4** Lineberger Comprehensive Cancer Center, University of North Carolina, Chapel Hill, NC, United States of America

☯ These authors contributed equally to this work.
‡ These authors also contributed equally to this work
* vjx3@cdc.gov

## Abstract

### Introduction

Around one-third of Americans reported they were unwilling to get a COVID-19 vaccine in April 2021. This focus group study aimed to provide insights on the factors contributing to unvaccinated adults' hesitancy or refusal to get vaccinated with COVID-19 vaccines.

### Method

Ipsos recruited 59 unvaccinated US adults who were vaccine hesitant (i.e., conflicted about or opposed to receiving a COVID-19 vaccination) using the Ipsos KnowledgePanel. Trained facilitators led a total of 10 focus groups via video-conference in March and April 2021. Two coders manually coded the data from each group using a coding frame based on the focus group discussion guide. The coding team collaborated in analyzing the data for key themes.

### Results

Data analysis of transcripts from the focus groups illuminated four main themes associated with COVID-19 vaccine hesitancy: lack of trust in experts and institutions; concern about the safety of COVID-19 vaccines; resistance towards prescriptive guidance and restrictions; and, despite personal reluctance or unwillingness to get vaccinated, acceptance of others getting vaccinated.

due to the small number of interviewees, there is significant potential for contextual clues within responses to reveal respondent identities. Thus, de-identified transcripts are available only upon request. Data requests should be sent to the CDC publishing coordinator (contact via publishingHD@cdc.gov).

**Funding:** This study was funded by the US Centers for Disease Control and Prevention, Ipsos 2021-50887. The content is solely the responsibility of the authors and does not necessarily represent official views of the funding organization.

**Competing interests:** Noel Brewer has served on paid advisory boards for Merck and received research grants from Merck and Pfizer. The remaining authors declare to have no conflicts of interest. This does not alter our adherence to PLOS ONE policies on sharing data and materials.

## Discussion

Vaccine confidence communication strategies should address individual concerns, describe the benefits of COVID-19 vaccination, and highlight evolving science using factural and neutral presentations of information to foster trust.

## Introduction

In the US, COVID-19 vaccines were authorized for use in December 2020 [1,2], and eligibility expanded to individuals 16 years of age and older by April 2021. Initial vaccine rollout challenges included difficulty scheduling appointments and insufficient supplies to meet the demand for vaccination [3]. But, for some subgroups, 'vaccine hesitancy,' defined by the World Health Organization's Behavioral and Social Drivers of Vaccination framework [4] as "a motivational state of being conflicted about, or opposed to, getting vaccinated," led to lower than expected rates of COVID-19 vaccine uptake. For example, a study from the Kaiser Family Foundation found that around 32% of Americans did not intend to get or were unsure about getting a COVID-19 vaccine in April 2021 [5].

US adults have been hesitant about COVID-19 vaccines for a variety of reasons ranging from questions about vaccine safety and effectiveness, trust in authorities, practical barriers, and other political and societal factors [6–8]. Moreover, research conducted early in the vaccine rollout period found that among those reluctant to receive COVID-19 vaccines, many expressed a willingness to get vaccinated if given additional safety and effectiveness information [9]. To develop successful strategies for promoting vaccine confidence, the US Centers for Disease Control and Prevention (CDC) contracted Ipsos Public Affairs (Ipsos) to coordinate 10 focus groups with US adults expressing vaccine hesitancy. This paper reports the focus group findings to improve our understanding of the reasons for vaccine hesitancy among select US adults and to inform the future development of strategies tailored to address these or similar concerns.

## Materials and methods

### Participants

Ipsos recruited adult participants from the Ipsos KnowledgePanel® [10]. CDC and Ipsos discussed eligibility criteria and selected participants for each focus group category using panel members' existing profile data on race, ethnicity, age, language preference, political ideology, and time zone. Including participants' time zone helped to ensure a wide variety of participants from different regions within the US. We screened participants for their likelihood of getting a COVID-19 vaccine using a questionnaire (S1 File). Eligible participants were those who indicated in the screening questions that they were unvaccinated and that they "definitely will not," probably will not," or were "not sure" if they would receive a COVID-19 vaccine. Eligible participants then consented to participate in focus groups.

### Instrument and procedure

The standardized focus group discussion guide drew from the Behavioral and Social Drivers of Vaccination (BeSD) Framework [4] and the Increasing Vaccination Model [11]. The BeSD framework identifies *thinking and feeling* and *social processes* as two domains most associated with vaccine hesitancy, or motivational conflict, and *practical issues* as a key domain that

moderates the relationship between vaccine hesitancy and vaccine uptake. ***Thinking and feeling*** was operationalized with questions that assessed information sources, rumors, and sentiments about COVID-19 vaccines and strategies to improve vaccine confidence. ***Social processes*** was operationalized with questions that assessed individual and perceived community attitudes toward COVID-19 vaccines. ***Practical issues*** was assessed with questions that asked about the impact of the COVID-19 pandemic on health and daily activities and barriers and enablers to COVID-19 vaccination in the community (S2 File). Participants who indicated that they "definitely will not" get, "probably will not" get, or were "not sure" if they would get a COVID-19 vaccine were considered vaccine hesitant.

Two informed consents were obtained from all focus group participants. In the online screening survey, participants were told the purpose of the study and the amount of time needed, possible risks and benefits of participation, and the incentive for participating (see S1 File). Participants who agreed to the online written informed consent were then provided additional verbal consent for the focus group discussion to be audio recorded.

Two moderators with doctoral degrees in social science facilitated focus group discussions which were conducted via video-conference between March 29 and April 8, 2021. Nine focus groups were conducted in English, and one was conducted in Spanish. Participants received a video-conference link and confirmed they had the necessary audio and video equipment before each meeting. Each focus group lasted approximately one hour, and each participant received a $75 cash-equivalent incentive for their participation. Transcripts and audio recordings for each group were documented and stored. Ethical review was conducted by the CDC. This activity was reviewed by the CDC and was conducted consistent with applicable federal law and CDC policy.

### Data analysis

A coding frame was developed based on the focus group discussion guide to code all focus group transcripts. A thematic analysis approach [12,13] was used to analyze the data collected during the focus group discussions. A team of seven researchers, all with degrees in social science or epidemiology, and supervised by a senior (PhD) author, conducted data analysis. Two coders, a primary coder and a secondary reviewer, examined the transcripts (with access to the recordings) to organize and categorize the data and resolve any discrepancies. The Spanish language transcript was translated into English and coded. All relevant transcript data were manually added to the coding frame by theme in Microsoft Excel and analyzed to find patterns of agreement and commonalities across groups. The analysis team conducted two team workshops to discuss preliminary findings and evaluate and interpret initial thematic results.

### Results

Table 1 lists participant characteristics. Participants' age ranged from 18 to 68 years (*Mean* = 43.6, *Standard Deviation* = 13.2), and there were 29 men and 30 women. Participants also varied by race, ethnicity, and political ideology.

Our analysis of the focus group data yielded four overarching themes: lack of trust in experts and institutions; concern about the safety of COVID-19 vaccines; resistance towards prescriptive guidance and restrictions; and, despite resistance towards vaccination for themselves, acceptance of others getting vaccinated. Additional details by theme follows.

### 1: Lack of trust in experts and institutions

One consistent attitude among vaccine-hesitant participants was a lack of trust in institutions or experts regarding COVID-19 vaccines. They were reluctant to believe the scientific and

**Table 1. Participant characteristics (*n* = 59).**

| Demographics | *n* (%) |
|---|---|
| **Age** | |
| 18–29 | 11 (19) |
| 30–49 | 28 (47) |
| 50–64 | 16 (27) |
| 65+ | 4 (7) |
| **Gender** | |
| Men | 29 (49) |
| Women | 30 (51) |
| **Race/Ethnicity** | |
| White, non-Hispanic | 23 (39) |
| Black, non-Hispanic | 17 (29) |
| 2+ Races or Other, non-Hispanic | 4 (7) |
| Hispanic | 15 (25) |
| **Education** | |
| No high school diploma or GED | 6 (10) |
| High school graduate or GED | 12 (20) |
| Some college or associate degree | 26 (44) |
| Bachelor's degree or higher | 15 (25) |
| **Political Ideology** | |
| Extremely conservative | 4 (7) |
| Conservative | 18 (31) |
| Slightly conservative | 3 (5) |
| Moderate | 19 (32) |
| Slightly liberal | 3 (5) |
| Liberal | 7 (12) |
| Extremely liberal | 0 (0) |
| Refused | 5 (8) |
| **Vaccine Hesitancy** | |
| Definitely will not get vaccinated | 16 (27) |
| Probably will not get vaccinated | 18 (31) |
| Not sure about getting vaccinated | 25 (42) |

medical community in general, and they resisted the advice of public health figures and their primary care providers who recommended COVID-19 vaccination. For example, when asked, "If your doctor said, 'Look, this thing is safe. It's the right thing to do,' would that motivate you?" a participant responded, *"No. They're not an immunologist. They're primary care. It's different, very different."* Some participants believed that their doctors were trustworthy but had only limited information: *"I think, for the most part, doctors and other people are trying to be truthful, but they don't know the complete information."*

Participants reported varying levels of trust associated with different sources. One participant described his level of trust in this way: *"Any elected official, definitely can't trust them. You definitely can't trust the media outlets."* Another participant also echoed the same viewpoint:

> *". . .you can't trust people in a position of power. It's not that power corrupts; it's that power attracts corruptible people. If we could trust that we are being given accurate information, that people had our best interest at heart, of course people would just follow it to save lives.*

*Why wouldn't you unless you're some nut or something? But the problem is that we can't trust them; we can't trust what they say."*

Distrust in expertise was prevalent across various racial and political groups, but especially among non-Hispanic Black participants. A participant in this group said, *"I feel like we can't trust the government. I feel like they're just going to tell us whatever we want to hear."* Another non-Hispanic Black participant shared:

*"We're extremely suspicious of the government because the government tends to repeatedly lie to the Black people about what is in a vaccine. They will come and say, 'Hey, this is going to cure you of this, or this is going to vaccinate you against this.' Meanwhile, they're conducting a study on you. So there is a huge suspicion on the government, and that is why a lot of Black people do not want to take this vaccine as well."*

In the Spanish-speaking Hispanic focus group, an additional trust-related concern was that undocumented immigrants might face issues if they attempt to get vaccinated:

*"I think that maybe people who are illegal, not everyone, but some people who are here illegally in the US may not do it out of fear. Even though they say that it's free and that there won't be any consequences or anything, people will probably avoid getting the vaccine for that very reason."*

Another participant within the Spanish-speaking Hispanic group agreed that this fear could be a barrier for some people:

*"I share my colleague's opinion about people who are foreigners here in this country. Sometimes, out of fear that they might be asked to show a document or something, maybe people don't want to get the vaccine for that reason. . . Sometimes, the information that circulates on social media says that they'll be detained. Others say that they won't. The myth that's going around social media or other forms of media makes people feel a little afraid. That creates a barrier for people."*

Participants were then asked who they thought could influence their choices, given that they did not accept the counsel of scientific or government experts. Some participants had other trusted sources. Hispanic participants regarded their faith organizations as potential sources of information. They agreed that faith organizations, such as churches, could influence them to get a COVID-19 vaccine, with one participant saying, *"[the churches were] important in our lives. Because if they weren't important, we wouldn't listen to them."* Another participant said that their church *"completely assure[d] everyone that the vaccine is very safe, that the vaccine is very appropriate."*

Since participants did not trust most sources of information, many felt that they could instead use their own judgment to determine the proper course of action. Participants generally had a great deal of confidence in their own ability to assess the risks of the pandemic, the role that social distancing and masks played in their safety, and the risks and opportunities associated with the vaccine. One participant presented the view that they know their health well enough to decide for themselves: *"But as far as the vaccine, no, I don't see any reason to get it. I've never had a flu shot, never had the flu, and I don't get sick."* Similarly, another participant felt that their immune system was enough to protect them:

*"If COVID has more than 99% recovery, I would just rather take the chances with my own immune system. . . There's just- our immune systems are fighting off germs on a daily basis, not just COVID, not just this new coronavirus. We're always fighting on something."*

Other participants discussed wanting to do their own research to understand the risk that COVID-19 posed to them, as well as if any of the three available vaccines were right for them:

*"I work for the community college doing a lot of data analysis, so digging into data is something I enjoy doing. I like to actually look at the numbers and see what is my actual risk. I don't want to take what I'm hearing in the news, so it's like, 'Let me look at it and see what my own risk is.' I read anything from the far-right that's out there that they said—well, say any number of things, and some of it sounds plausible."*

Similarly, another participant explained how they looked at peer-reviewed literature to do their own research:

*"I typically take a look at science articles on PubMed. In particular, one thing that I notice is that I take a look at the old research, say from the 1990s, all the way through, say, the 2010s, and I take a look at some of the modern research that's coming."*

## 2: Concern about the safety of COVID-19 vaccines

Vaccine hesitant participants questioned the long-term safety of COVID-19 vaccines—particularly the Moderna and Pfizer vaccines—given their uncertainty about the novel technology used to develop them. Some indicated that the long-term consequences are unknown; therefore, scientists and medical experts cannot accurately conclude that the vaccines are safe. One participant described the perceived lack of information and transparency regarding vaccine safety:

*". . .I'm very technical in nature, so give me the facts to show me. Back up your opinion. If you have facts, then show them to me instead of just saying, 'I don't believe they will have, in our opinion, they don't have long-term side effects.' Show me the facts of it. Back it up with facts. And once again, with Moderna and Pfizer, with the new technology and the new way of making it, who knows what those long-term effects are?"*

Another participant offered the following perspective:

*"Nobody can convince me. My doctor has tried to convince me. She has told me to come into her office, and she'll try to give it to me. No. . . I have to see proof that people are OK five years down the road. They didn't grow a third eye, or all of a sudden, they didn't get cancer or something. Something did not go wrong. I need to see proof."*

Additionally, some participants believed that the public was being manipulated by powerful actors when it came to vaccine safety: *"I can say I don't have much trust in whatever I'm listening [to], because I see [a] pharmaceutical company is very powerful. They have very powerful tools to communicate; they can manipulate people very easily."*

Some groups expressed doubts about the safety of the vaccines based on perceptions of the speed of their development: *"[I'm] a little skeptical because it happened so fast. Because usually, [a] vaccine takes so many years to develop, and this happened so quickly. I'm really skeptical about it. So right now, I'm just waiting to see how things go."*

However, other participants had more specific concerns about the possible effects on pregnancy. One participant offered the following:

*"It's awesome that they've been able to fast track it so much, but just not having as much of the research and testing hasn't been able to be completed, not knowing. They say it's safe for pregnancy, but they haven't done the long-term testing on it."*

Another participant said that while it was great that the vaccine was generally available, they still did not want it due to long-term safety concerns:

*"I personally don't want it yet, because as a young adult, I don't know what the long-term side effects could be. Especially being a woman, I don't know how it would affect me if I wanted to have children in the future."*

### 3: Resistance toward prescriptive guidance and restrictions

Participants expressed dismay with what they saw as an overly prescriptive tone to the guidance they received about COVID-19 prevention practices. Some participants wanted to receive information about vaccine safety and efficacy from experts but also wanted opportunities to make up their own minds about getting vaccinated. One participant explained how they were already doing their part in protecting others, and they felt that the vaccine should be a personal choice and should not be prescribed by experts:

*"I feel like the social distancing, the wearing of the mask, the washing my hands, that's my duty to other people. In terms of the vaccine, I don't think. . . I almost feel like that's kind of manipulative. . . because we had doctors, we've had pastors, we've had community leaders, and everybody's like 'get the vaccine, it helps other people,' and I'm just like, at some point, you got to understand there's a free world aspect of this."*

Some participants also felt pressured by their communities to get vaccinated but were not yet convinced: *"There's a lot of pressure, especially in the Black community, to get vaccinated, and I hope that all is well, just I don't want to take a risk on something."* Additionally, despite knowing that certain communities (such as Black, Hispanic, American Indian, and Alaska Native communities) were disproportionately affected by higher rates of illness and severe disease, many respondents from these communities were still not inclined to get vaccinated. A non-Hispanic Black participant said the following:

*"I'm always trying to find anything about it that will make me more positive toward it, but I'm not, and I know a lot of people in my community, friends and family, there's a lot of fear behind it, because they say the African American community is more susceptible, and in more debt, and then they come up with this vaccine, and it's like a fear, if I don't get it, somebody in my family will die. The African American community around me, a lot of them are very fearful that if they don't get it, something bad will happen, and I don't like the fact that they're putting it out like that, and it's making, it's a source of fear among some people."*

Additionally, participants expressed frustration and questioned the purpose of the vaccines if they were still encouraged or required to engage in nonpharmaceutical interventions (NPIs) —such as continued mask-wearing, social distancing, or capacity limitations at restaurants and other locations—despite receiving the vaccines:

*"You still have to wear your mask, but you've gotten vaccinated, so you mean to tell me the vaccine hasn't protected you? What's going on there? There [are] just too many unknown questions, it seems, that are still out there about it that I don't know that there [are] answers for either."*

Inconsistent guidance on masks from the CDC and the World Health Organization early in the pandemic was also a source of frustration:

*"'Don't wear a mask because it's not going to help,' and that was based off [of] studies back in the 90s and the 2000s. Then they said, 'wear a mask,' and then they started saying, 'wear a double mask and even a triple mask.' They're just not very consistent with their messaging."*

The awareness of varied prevention guidance within and across states underlined participants' frustration with mixed or changing messaging. For example, some participants in states with stricter mitigation efforts expressed frustration that they remained subjected to some restrictions when other states eliminated restrictions and 'opened up.' However, when participants discussed whether the loosening of restrictions served as motivating factor for vaccination, most participants indicated it was not an influential factor on their decision.

In addition to their overall resistance to prescriptive guidance, participants also feared possible further restrictions, such as community settings requiring "vaccine passports," where access to certain venues could be dependent on providing proof of having received a COVID-19 vaccine. Vaccine passports were perceived as a basic infringement on individual liberty. None of the participants spoke in favor of a vaccine passport. Several participants worried that vaccine passports would become a reality and saw them as the gravest risk posed to personal liberty and hailed various institutions, including the Constitution, as offering protection from mandated vaccination. One participant said that the potential requirement of vaccine passports was tyrannical:

*"But I also don't think the recent stuff has been coming out in the news, a vaccine passport. I disagree with people being fought and try- some attempt to try to force people to take a vaccine and then that he was in the news talking about, 'Oh, we need to dangle the carrot and not open things up until people take this vaccine.' And I don't like that. I don't like that threat of 'Oh, well. We not going to- businesses ain't going to let you in unless you show your ID card.' We're not going to let- I don't like any type of tyranny like that."*

## 4: Acceptance of others getting vaccinated

While nearly all participants held strong views about not getting vaccinated themselves, some were accepting of others' decisions to get vaccinated or even helped family members, such as parents, grandparents, or children, get vaccinated:

*"I've spoken with family members, though, with my mom because they're not with me. They're in another country. I try to find a way when I talk to my mom sometimes because she's older. She suffers from many chronic diseases. In fact, she's very aware that she's going to get vaccinated. I'm always sharing information with her, more than anything. Not telling her to get vaccinated, but I do share information with her about what I've heard here about the vaccines and the effect that they have. The good thing is getting vaccinated and everything. She tells me, 'Yes, dear, as soon as there's an opportunity, I'll get vaccinated.'"*

Even if some participants opposed family members' choices to get vaccinated, it was seemingly "softer" opposition. For instance, one participant had no plans to be vaccinated himself, but offered only limited resistance when his wife got vaccinated and made an appointment for their son: *"She's already got an appointment for him for next weekend. Over the next month, if you want it, you should be able to get it in this area."* Among the participants who expressed their willingness to help others find and obtain vaccines, the majority identified as liberal-leaning.

Some participants stated that they were willing to get vaccinated if people they trusted did not suffer negative consequences from getting a COVID-19 vaccine. One woman described that she was waiting for her parents to get their second dose of the vaccine before she decided if she would get it as well:

*"But my mom and dad also have COVID, and they received their first dosage two weeks ago. I'm just waiting for them to get the second dosage and then take it from there and make sure that I'm safe to be able to get my vaccine."*

## Discussion

This qualitative study investigated perceptions of COVID-19 vaccines' safety and effectiveness, vaccination guidance, and social norms regarding vaccination among vaccine-hesitant US adults. Four themes were identified: distrust of experts and institutions regarding COVID-19 vaccines; worries about the long-term effects and safety of COVID-19 vaccines; resistance toward prescriptive guidance and pressure regarding vaccination and social restrictions; and acceptance, and in some cases encouragement, of others getting vaccinated. The elicited themes primarily centered around the two domains from the BeSD Framework most associated with vaccine hesitancy, or motivational conflict ***thinking and feeling issues***—notably, distrust of experts and institutions promoting COVID-19 vaccines and worries about the long-term effects and safety of COVID-19 vaccines, ***social processes***—notably, resistance toward prescriptive guidance, social restrictions, and social pressure regarding vaccination, but also acceptance, and in some cases encouragement, of others getting vaccinated. These findings underscore the important role of considering intra- and inter-personal level factors on COVID-19 vaccination uptake above and beyond access, availability, and other "practical" issues.

### Strategies to increase public confidence in vaccines

The focus group results suggest that many people who are hesitant to get a COVID-19 vaccine had concerns regarding vaccine safety. Messages about COVID-19 vaccines that provide clear, fact-based responses to the types of questions people have about topics like the vaccine approval process, the low risk of short- or long-term side effects, and the interrelationship between vaccination and NPIs such as social distancing, wearing face masks, and frequent handwashing may help improve confidence in COVID-19 vaccine safety and effectiveness.

Consistent with some reports on mitigation behaviors, some participants felt that they were already doing their part for their community by practicing social distancing and mask-wearing, and thus felt that vaccination should relegated to a personal decision rather than framed as a social responsibility. However, strategies to increase vaccine confidence can also focus on the benefits of vaccination for oneself and one's family and friends. In line with findings of previous studies on HPV vaccination and physical activities [14,15], an effective approach for COVID-19 vaccination could be to reframe vaccination from societal pressure to an

empowering behavior to protect family, friends, and others in their immediate social networks [16,17]. In addition, strategies should highlight that the risk of side effects from COVID-19 vaccination is much lower than the risk of severe symptoms from COVID-19 [18,19].

Furthermore, some participants expressed that they did not understand how vaccination was more effective than the NPI behaviors they were already practicing, or why NPIs were still needed for vaccinated persons. The focus group findings suggest that information should be provided that emphasizes the complementary role of NPIs to the essential role vaccines play in preventing severe illness and death [20–22].

Trusted community leaders should also be identified to avoid a "one size fits all" messaging approach [15,23]. Therefore, to improve attitudes towards COVID-19 vaccines, appropriate messengers who appeal to different subgroups should be identified and enlisted to address people's concerns about the vaccines' long-term safety and potential risks, particularly as these concerns can be entrenched [24–28]. In engaging non-Hispanic Black persons, recognizing past public health failures by the government and medical communities may be a positive step in building trust [16,26,27,29,30]. After acknowledging past failures, trusted messengers could then listen to ongoing vaccination concerns [31,32] and engage trusted community leaders to provide responsive, tailored information to address the concerns [33]. Future research in this area should explore the complex dynamics between communities and trusted messengers, particularly in instances where messengers are recognized as trusted but health recommendations are not accepted.

## Strengthening social norms to get vaccinated

Except for individuals in the most vaccine-hesitant focus group, participants were less resistant to others getting a COVID-19 vaccine, and some even engaged in activities that facilitated others getting vaccinated. This may suggest that people are trying to balance the potential risks and benefits and believe that when someone is more vulnerable to COVID-19, the perceived risks of short- or long-term side effects following vaccination are outweighed by the protections that vaccines offer. That, in turn, suggests that presenting relative risks (i.e., severe symptoms of COVID-19 vs. severe side effects of the vaccine) could influence some hesitant individuals.

Alternatively, willingness to help others can be explained by the extent to which a person believes that vaccination poses a social contract with which others should comply [34–37]. One technique for increasing vaccination rates could be to use the stories of vaccinated individuals to encourage unvaccinated people in their families and communities to receive COVID-19 vaccines [38–41]. Public health strategies could also employ peoples' vaccination narratives, mainly why they decided to get vaccinated and the effect that it has had, including acknowledgment of their initial fears and side effects [19,40–43].

## Strengthen practical facilitators

Overall, practical issues around vaccination associated with vaccine uptake, such as difficulty traveling to vaccination facilities or scheduling appointments, did not emerge as serious concerns among focus group participants because most participants had not attempted to get vaccinated, which would have revealed potential barriers. One exception to this was that Spanish-speaking Hispanic participants shared concerns about vaccine processes being used to identify undocumented immigrants. As such, informing those who work with undocumented immigrants about this concern is critical so that staff can offer reassurances about the separation of vaccination from immigration enforcement activities and can learn about their clients' other potential barriers or concerns [43].

## Limitations

This study has four primary limitations. First, while the focus group sample consisted of participants from various races, ethnicities, ages, genders, and political ideologies, the average age of the sample was over 40 years old; thus, the data may not necessarily reflect the opinions of younger people who may, for instance, receive vaccination information through different channels. Future focus group studies are needed to identify appropriate strategies focusing on younger populations. Second, the total number of participants and number of persons in different racial, ethnic, and political ideology groups was small, and thus, caution should be taken in interpreting the findings or generalizing to larger subgroups. This is particularly true for speakers of languages other than English or Spanish. Third, due to the COVID-19 restrictions, all focus groups were conducted online using video-conference. Individuals from communities without reliable access to the internet or other low-resourced individuals may be underrepresented in this study. Finally, the data for this study were elicited from focus group discussions, which may not be representative or generalizable to broader populations. Additionally, due to small subpopulation sample sizes, in-depth exploration of themes among any one group was not possible. While our findings can be used to inform strategies to increase vaccine confidence, more qualitative and quantitative survey research as well as evaluation of programs implemented to increase vaccine confidence is needed to determine the effectiveness of these strategies.

## Conclusions

Participants in this study of COVID-19 vaccine hesitancy among US adults offered three explanations for their reluctance to get vaccinated: distrust in public health experts and institutions; concerns about the long-term consequences and safety of COVID-19 vaccines; and resistance to prescriptive guidance and frustration with social restrictions. However, despite their reluctance to get a vaccine for themselves, some participants indicated a willingness to help others get vaccinated. Public health practitioners at all levels can consider using these findings to implement strategies and messages to strengthen confidence in COVID-19 vaccines, particularly when information about a novel disease and vaccine is complex and rapidly changing.

## Supporting information

**S1 File. Screening questionnaire.**
(DOCX)

**S2 File. Discussion questions.**
(DOCX)

## Author Contributions

**Conceptualization:** Neetu Abad, Seth D. Messinger, Qian Huang, Megan A. Hendrich, Helen Fisun, Zachary Lewis, Elisabeth Wilhelm, Brittney Baack, Kimberly E. Bonner, Rosemarie Kobau, Noel T. Brewer.

**Data curation:** Neetu Abad.

**Formal analysis:** Neetu Abad, Seth D. Messinger, Nataly Johanson, Noel T. Brewer.

**Investigation:** Seth D. Messinger, Qian Huang, Megan A. Hendrich.

**Methodology:** Neetu Abad, Seth D. Messinger, Qian Huang, Megan A. Hendrich, Nataly Johanson, Helen Fisun, Zachary Lewis, Elisabeth Wilhelm, Brittney Baack, Kimberly E. Bonner, Rosemarie Kobau, Noel T. Brewer.

**Project administration:** Neetu Abad, Megan A. Hendrich, Helen Fisun, Zachary Lewis.

**Supervision:** Neetu Abad, Megan A. Hendrich, Helen Fisun, Noel T. Brewer.

**Validation:** Seth D. Messinger, Qian Huang, Noel T. Brewer.

**Writing – original draft:** Neetu Abad, Seth D. Messinger, Qian Huang, Megan A. Hendrich, Nataly Johanson, Noel T. Brewer.

**Writing – review & editing:** Neetu Abad, Seth D. Messinger, Qian Huang, Megan A. Hendrich, Zachary Lewis, Elisabeth Wilhelm, Brittney Baack, Kimberly E. Bonner, Rosemarie Kobau, Noel T. Brewer.

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
