## [Decision Letter · Decision Letter 0]

4 Aug 2022

PONE-D-22-12427A qualitative study of behavioral and social drivers of COVID-19 vaccine confidence and uptake among unvaccinated Americans in the US April-May 2021PLOS ONE

Dear Dr. Abad,

Thank you for submitting your manuscript to PLOS ONE. After careful consideration, we feel that it has merit but does not fully meet PLOS ONE’s publication criteria as it currently stands. Therefore, we invite you to submit a revised version of the manuscript that addresses the points raised during the review process.

Both reviewers as well as myself find merit in the submission. I think both Reviewer 1 and 2 make a number of excellent suggestions that improve the readability and clarity of the text and results. All the proposed changes seem doable and I urge you to consider making these changes to the best of your ability.

We look forward to receiving your revised manuscript.

Kind regards,

Lorien Shana Jasny

Academic Editor

PLOS ONE

Journal Requirements:

 "This study was funded by the US Centers for Disease Control and Prevention, Ipsos 2021-50887"

   "Noel Brewer has served on paid advisory boards for Merck and received research grants from Merck and Pfizer. The remaining authors declare to have no conflicts of interest."

Reviewers' comments:

Reviewer's Responses to Questions

**Comments to the Author**

1. Is the manuscript technically sound, and do the data support the conclusions?

Reviewer #1: Yes

Reviewer #2: Partly

2. Has the statistical analysis been performed appropriately and rigorously? 

Reviewer #1: N/A

Reviewer #2: N/A

3. Have the authors made all data underlying the findings in their manuscript fully available?

Reviewer #1: Yes

Reviewer #2: Yes

4. Is the manuscript presented in an intelligible fashion and written in standard English?

Reviewer #1: Yes

Reviewer #2: Yes

5. Review Comments to the Author

Reviewer #1: Summary of the research

Manuscript reference: PONE-D-22-12427

Title: A qualitative study of behavioral and social drivers of COVID-19 vaccine confidence and uptake among unvaccinated Americans in the US April-May 2021

General comments:

The qualitative study aimed to improve the understanding of the reasons for vaccine hesitancy among US adults.

A purposive sample of adults from different demographics across the US that met the study selection criteria were selected to participate in online focus group discussions using a discussion guide based on the Ipsos and CDC collaborated standardized focus group discussion guide for COVID-19 related issues. The transcribed data was coded using a coding frame based on this discussion guide and analyzed thematically.

The efforts of the authors conduct and report the findings of this research is commendable.

Major comments:

I have no major comments, rather an important suggestion that I would want the authors to consider in future for similar studies.

In the data analysis section of the Methods section, it was stated that “A coding frame was developed based on the focus group discussion guide to code all focus group transcripts”. The discussion guide seems to be rather prescriptive and restrictive, and this may have affected the discussion; which in turn is reflected in the results reported.

Perhaps, in a future work, it can be made less so, and questions and probes framed in such a way that would elicit remedial and solution-oriented responses from the participants (even though they were all classified as vaccine-hesitant, not all of them were in the “definitely not” subgroup. Some were in the “probably will not,” or “not sure” subgroup, these could have provided such responses). For example, in Appendix B, under “E. Strategies to Improve Vaccine Confidence in the Community”, questions such as what do you think will improve vaccine confidence in your community? What do you think are the strategies currently in place to improve vaccine confidence in your community? Are they effective in your opinion? How do you think they can be improved? This types of questions can be included [in place of], or at least, in addition to the ones currently asked.

Minor comments:

Abstract: The abstract is succinct and well written.

However, as stated, if around a third of Americans reported their unwillingness to receive the COVID-19 vaccine in May 2021, and the focus group discussion took place in March-April 2021; I will advise that the accuracy of the dates be checked. If the dates are accurate, then the statement “This focus group study aims to provide insights on the factors contributing to unvaccinated adults’ vaccine hesitancy… may need to be rephrases as the investigations were conducted before May 2021.

Also, I suggest that the phrase “as in” in the methods section of the abstract be replaced with “that is” or “meaning” or even “i.e.”

Introduction: Insert the word ‘program’ or ‘period’ (or whichever word is applicable) between the words ‘rollout’ and ‘found’.

Results: I will suggest that participants’ quotes be italicized if this complies with the journals’ stipulated writing format.

Reviewer #2: During the pandemic, understanding where people are coming from—e.g., their experiences with COVID-19; concerns and hopes around COVID-19 vaccines; and the realities of their health and living situations, and access to care—is foundational to developing public health interventions that work. This study provides in people’s own words, reasons for delaying or foregoing COVID-19 vaccination. Such research can inform efforts to improve vaccine delivery and communication strategies, reaching as many people as possible.

Suggestions for improving the manuscript follow below:

INTRODUCTION

Provide more context and detail regarding the vaccine rollout – Based on the current prose, a reader could walk away thinking that COVID-19 vaccines only became available for U.S. adults in April 2021. While initially limited by supply and prioritization schemes, the first vaccines began distribution in mid/late December. By the time of the data collection (March 29-April 8), many factors were already shaping people’s impressions about COVID-19 vaccines and where they fit in their lives, including early reports about potential adverse effects (e.g., allergic reactions, heart failure).

Strengthen the use of source(s):

• Clear up confusion about source dated 2020 to support statement referring to 2021 (p 3, para 1)

• Provide source for statement re: scheduling challenges and insufficient supplies (p 3, para 1)

Delve more deeply into the “vaccine hesitancy” literature – The concept of “vaccine hesitancy” is central to the study. Yet, the researchers abruptly distill it down to “the reluctance or refusal to vaccinate. As a result, they and their readers do not benefit from the more nuanced treatment by the WHO Strategic Advisory Group of Experts (SAGE) Working Group on Vaccine Hesitancy: “[V]accine hesitancy refers to delay in acceptance or refusal of vaccination despite availability of vaccination services. Vaccine hesitancy is complex and context specific, varying across time, place and vaccines. It is influenced by factors such as complacency, convenience and confidence.” A consequence is the elision of personal attitudes toward vaccines with practical circumstances that enable/inhibit access to vaccines.

State more clearly the purpose of the study – Was it to test specific messages or to uncover reasons for vaccine hesitancy that could then inform the development of specific messages? (p 3, para 2)

METHODS

Provide rationale for your 5 selected discussion topics, including whether they were informed by other empirical studies or a particular theoretical framework. Note that these topics also drove the development of the coding frame, so it is important to share the reasoning.

Describe the characteristics/credentials of the research team (e.g., are the coders trained in thematic analysis?). The authors may wish to review the COREQ (COnsolidated criteria for REporting Qualitative research) Checklist to aid them in strengthening the write-up of the study. Allison Tong, Peter Sainsbury, Jonathan Craig, Consolidated criteria for reporting qualitative research (COREQ): a 32-item checklist for interviews and focus groups, International Journal for Quality in Health Care, Volume 19, Issue 6, December 2007, Pages 349–357, https://doi.org/10.1093/intqhc/mzm042

Clarify how the data were analyzed – e.g., What software, if any, was used to manage the data?

RESULTS

Consider the socioeconomic status of research subjects – The participants skew toward more highly educated persons (and potentially those with greater economic means). Such a point is important; among more highly resourced people, issues of impaired access (e.g., can’t leave an hourly wage job, lacks transportation, lacks childcare) may less salient.

Explain the idea of “vaccine passports” (p 13, last para)

Confirm how the data analysis led to this finding – “Liberal-leaning participants particularly expressed their willingness to help others find and obtain vaccines.” What is the evidence to support this statement? (p 15, para 1)

DISCUSSION

Expand the discussion of the “Increasing Vaccination Model” and show, not just tell, the connection between study results and the model. (pp 15-16)

Define “vaccine hesitancy spectrum” (p 10, para 1) – Neither the introduction nor the discussion includes an explanation of this framework.

Relate finding to the existing COVID-19 vaccine hesitancy literature – Does the study’s findings and recommendations resonate or not with other relevant research? As of now, a Pub Med search for “COVID-19 vaccine hesitancy United States” produces 462 results, and the article’s bibliography only includes ~8 other studies. The authors should embed their work much more thoroughly in the larger research. For example:

• Are there other studies that could strengthen this proposal (p 16, para 1) – “Messages about COVID-19 vaccines that provide clear, fact-based responses to the types of questions people have about topics like the vaccine approval process, the low risk of short- or long-term side effects, and the interrelationship between vaccination and NPIs such as social distancing, wearing face masks, and frequent and washing may help improve confidence in COVID-19 vaccine safety and effectiveness.”

• Do any other COVID-19 vaccine hesitancy studies have similar finding (p 16, para 2) – “[S]ome participants felt that they were already doing their part for their community by practicing social distancing and mask-wearing and thus felt that vaccination should be a personal decision rather than one based on social responsibility.”

Refrain from over-generalizing – “These findings indicate that societal pressure or messaging based on protecting one’s community may not be effective in increasing COVID-19 vaccination.” (p 16, para 2) Is this always true? What might other COVID-19 vaccine related (and other) research suggest?

Resolve the apparent contradiction between the following claims (p 16, para 2-3) –

• “These findings indicate that societal pressure or messaging based on protecting one’s community may not be effective in increasing COVID-19 vaccination.”

• “…an effective approach for COVID-19 vaccination could be to encourage vaccination behavior that aligns with protecting themselves as well as their family, friends, and others in their immediate social networks.”

Clarify the logic in the sequence of your ideas –

• What is the logical step missing between these 2 sentences (p 17, para 1): “Furthermore, some participants expressed that they did not understand how vaccination was more effective than the NPI behaviors they were already practicing or why NPIs were still needed for vaccinated persons. The focus group findings suggest that information should be provided that emphasizes the essential role vaccines play in preventing severe illness and death (Griffith et al., 2010; Haldane et al., 2019). “

• How does this order of ideas make sense (p 17, para 2): “In engaging non-Hispanic Black persons, recognizing past public health failures by the government and medical communities may be a positive step in building trust (Momplaisir et al., 2021; Pichon et al., 2012). Such messengers could emphasize the relatively low incidence of severe side effects identified even after hundreds of millions of COVID-19 vaccine doses have been administered in the US and globally (Gee et al., 2021).

6. PLOS authors have the option to publish the peer review history of their article (what does this mean?). If published, this will include your full peer review and any attached files.

Reviewer #1: **Yes: **Elizabeth O. Oduwole

Reviewer #2: No

---

## [Author Response · Author response to Decision Letter 0]

3 Jan 2023

To Dr. Lorien Shana Jasny,

Thank you for careful review of this manuscript. We have addressed each of the requirements denoted below and in the requested areas of the manuscript, cover letter and submission portal. 

Journal Requirements:

We have revised the manuscript and author affiliations in accordance with these style requirements.

 "This study was funded by the US Centers for Disease Control and Prevention, Ipsos 2021-50887"

We have updated the funding statement in the cover letter, as per CDC guidance for publications: 

“This study was funded by the US Centers for Disease Control and Prevention, Ipsos 2021-50887. The content is solely the responsibility of the authors and does not necessarily represent official views of the funding organization.”

 "Noel Brewer has served on paid advisory boards for Merck and received research grants from Merck and Pfizer. The remaining authors declare to have no conflicts of interest."

We have updated the competing interests section in the cover letter, as follows:

“Noel Brewer has served on paid advisory boards for Merck and received research grants from Merck and Pfizer. The remaining authors declare to have no conflicts of interest. This does not alter our adherence to PLOS ONE policies on sharing data and materials.“ 

We have updated our data availability statement, as follows: 

“The authors have shared summary data tables as allowed by the ethical clearance procedures of the CDC. However, due to the small number of interviewees, there is significant potential for contextual clues within responses to reveal respondent identities. Thus de-identified transcripts are available only upon request. Data requests should be fielded to CDC publishing coordinator: PublishingHD@cdc.gov.”

We have updated the ethics statement to clarify that the CDC conducted the ethical review, and we have retained the CDC’s required language regarding ethical review, as follows (lines 130-132): 

“Ethical review was conducted by the CDC. This activity was reviewed by the CDC and was conducted consistent with applicable federal law and CDC policy. ”

Thanks you for this suggestion. We have also included an erratum to the interim Advisory Committee on Immunization Practices’ Interim Recommendation for Use of Pfizer-BioNTech COVID-19 Vaccine:

Oliver SE, Gargano JW, Marin M, Wallace M, Curran KG, Chamberland M, et al. The Advisory Committee on Immunization Practices' Interim Recommendation for Use of Pfizer-BioNTech COVID-19 Vaccine - United States, December 2020. MMWR Morb Mortal Wkly Rep. 2020;69(50):1922-4. Epub 20201218. doi: 10.15585/mmwr.mm6950e2. PubMed PMID: 33332292; PubMed Central PMCID: PMCPMC7745957.

Added reference:

Erratum: Vol. 69, No. 50. MMWR Morb Mortal Wkly Rep 2021;70:144. DOI: http://dx.doi.org/10.15585/mmwr.mm7004a5

Manuscript reference: PONE-D-22-12427

Title: A qualitative study of behavioral and social drivers of COVID-19 vaccine confidence and uptake among unvaccinated Americans in the US April-May 2021

We would like to extend our thanks to these reviewers for their careful review of this manuscript as well as their constructive suggestions. We have included each reviewer’s comment, along with a response. Any changes to manuscript text in response to reviewer feedback is also included in our response for reference.

Reviewer 1: I have no major comments, rather an important suggestion that I would want the authors to consider in future for similar studies. In the data analysis section of the Methods section, it was stated that “A coding frame was developed based on the focus group discussion guide to code all focus group transcripts”. The discussion guide seems to be rather prescriptive and restrictive, and this may have affected the discussion; which in turn is reflected in the results reported.

Perhaps, in a future work, it can be made less so, and questions and probes framed in such a way that would elicit remedial and solution-oriented responses from the participants (even though they were all classified as vaccine-hesitant, not all of them were in the “definitely not” subgroup. Some were in the “probably will not,” or “not sure” subgroup, these could have provided such responses). For example, in Appendix B, under “E. Strategies to Improve Vaccine Confidence in the Community”, questions such as what do you think will improve vaccine confidence in your community? What do you think are the strategies currently in place to improve vaccine confidence in your community? Are they effective in your opinion? How do you think they can be improved? This types of questions can be included [in place of], or at least, in addition to the ones currently asked.

Authors’ response: This suggestion is well-noted and will be considered for future studies.

Reviewer 1: Abstract: The abstract is succinct and well written. However, as stated, if around a third of Americans reported their unwillingness to receive the COVID-19 vaccine in May 2021, and the focus group discussion took place in March-April 2021; I will advise that the accuracy of the dates be checked. If the dates are accurate, then the statement “This focus group study aims to provide insights on the factors contributing to unvaccinated adults’ vaccine hesitancy… may need to be rephrases as the investigations were conducted before May 2021.

Authors’ response: We have updated the abstract and introduction to highlight the Kaiser Family Foundation data from April 2021.

Reviewer 1: Also, I suggest that the phrase “as in” in the methods section of the abstract be replaced with “that is” or “meaning” or even “i.e.”

Introduction: Insert the word ‘program’ or ‘period’ (or whichever word is applicable) between the words ‘rollout’ and ‘found’.

Results: I will suggest that participants’ quotes be italicized if this complies with the journals’ stipulated writing format.

Authors’ response: Thank you. We have made the recommended changes. 

Reviewer 2: Provide more context and detail regarding the vaccine rollout – Based on the current prose, a reader could walk away thinking that COVID-19 vaccines only became available for U.S. adults in April 2021. While initially limited by supply and prioritization schemes, the first vaccines began distribution in mid/late December. By the time of the data collection (March 29-April 8), many factors were already shaping people’s impressions about COVID-19 vaccines and where they fit in their lives, including early reports about potential adverse effects (e.g., allergic reactions, heart failure).

Strengthen the use of source(s):

• Clear up confusion about source dated 2020 to support statement referring to 2021 (p 3, para 1)

• Provide source for statement re: scheduling challenges and insufficient supplies (p 3, para 1)

Authors’ response: Thank you. We have made the recommended changes.

Reviewer 2: Delve more deeply into the “vaccine hesitancy” literature – The concept of “vaccine hesitancy” is central to the study. Yet, the researchers abruptly distill it down to “the reluctance or refusal to vaccinate. As a result, they and their readers do not benefit from the more nuanced treatment by the WHO Strategic Advisory Group of Experts (SAGE) Working Group on Vaccine Hesitancy: “[V]accine hesitancy refers to delay in acceptance or refusal of vaccination despite availability of vaccination services. Vaccine hesitancy is complex and context specific, varying across time, place and vaccines. It is influenced by factors such as complacency, convenience and confidence.” A consequence is the elision of personal attitudes toward vaccines with practical circumstances that enable/inhibit access to vaccines.

Authors’ response: Thank you for highlighting the importance of this literature. We have updated the text to position “vaccine hesitancy” within the “Motivation” domain of the WHO’s recently published guidance of the Behavioral and Social Drivers of Vaccination. The manuscript text has been updated as follows: ”In addition, ‘vaccine hesitancy’ , defined by the World Health Organization’s Behavioral and Social Drivers of Vaccination framework (WHO 2022) as “a motivational state of being conflicted about, or opposed to, getting vaccinated”, led to lower than expected rates of COVID-19 vaccine uptake.”

We have updated the manuscript text (lines 27-28, 51-53) as follows:

“But, for some subgroups, ‘vaccine hesitancy,’ defined by the World Health Organization’s Behavioral and Social Drivers of Vaccination framework [4] as “a motivational state of being conflicted about, or opposed to, getting vaccinated,” led to lower than expected rates of COVID-19 vaccine uptake.”

The extract below from the WHO illustrates that this updated definition supersedes the SAGE definition of vaccine hesitancy from 2014. 

“’Vaccine hesitancy’ is part of the Motivation domain and defined as a motivational state of being conflicted about, or opposed to, getting vaccinated; this includes intentions and willingness. This definition replaces that given by SAGE in 2014, where vaccine hesitancy was defined as a delay in acceptance or refusal of vaccination despite availability of vaccination services. The new definition recognizes hesitancy as an intention or motivation and is separate to the resulting behaviour. This enables behaviours and their many other influences to be better understood and measured separately.”

State more clearly the purpose of the study – Was it to test specific messages or to uncover reasons for vaccine hesitancy that could then inform the development of specific messages? (p 3, para 2)

Authors’ response: We have clarified the aims statement as follows (lines 79-82): 

“This paper reports the focus group findings to improve our understanding of the reasons for vaccine hesitancy among select US adults and to inform the future development of strategies tailored to address these or similar concerns.” 

METHODS

Provide rationale for your 5 selected discussion topics, including whether they were informed by other empirical studies or a particular theoretical framework. Note that these topics also drove the development of the coding frame, so it is important to share the reasoning.

Authors’ response: We have updated the methods section to directly link between the Behavioral and Social Drivers of Vaccination Framework and the five selected discussion topics. The text now reads as follows (lines 97-115):

The standardized focus group discussion guide drew from the Behavioral and Social Drivers of Vaccination (BeSD) Framework [4] and the Increasing Vaccination Model [11]. The BeSD framework identifies thinking and feeling, and social processes as two domains most associated with vaccine hesitancy, or motivational conflict, and practical issues as a key domain that moderates the relationship between vaccine hesitancy and vaccine uptake. Thinking and feeling was operationalized with questions that assessed information sources, rumors, and sentiments about COVID-19 vaccines and strategies to improve vaccine confidence. Social processes was operationalized with questions that assessed individual and perceived community attitudes toward COVID-19 vaccines. Practical issues was assessed with questions that asked about the impact of the COVID-19 pandemic on health and daily activities and barriers and enablers to COVID-19 vaccination in the community (S2 File). 

Describe the characteristics/credentials of the research team (e.g., are the coders trained in thematic analysis?). The authors may wish to review the COREQ (COnsolidated criteria for REporting Qualitative research) Checklist to aid them in strengthening the write-up of the study. Allison Tong, Peter Sainsbury, Jonathan Craig, Consolidated criteria for reporting qualitative research (COREQ): a 32-item checklist for interviews and focus groups, International Journal for Quality in Health Care, Volume 19, Issue 6, December 2007, Pages 349–357, https://doi.org/10.1093/intqhc/mzm042

Author’s Response: Two moderators with doctoral degrees in social science facilitated focus group discussions. One of these social scientists supervised a team of seven researchers who conducted thematic data analysis. All researchers had at least a college education and training in social science or epidemiology, along with specific training, guidance, and supportive supervision to conduct the thematic analysis for this study.

Clarify how the data were analyzed – e.g., What software, if any, was used to manage the data?

Author’s Response: Microsoft Excel was used to analyze the data

RESULTS

Consider the socioeconomic status of research subjects – The participants skew toward more highly educated persons (and potentially those with greater economic means). Such a point is important; among more highly resourced people, issues of impaired access (e.g., can’t leave an hourly wage job, lacks transportation, lacks childcare) may less salient.

Authors’ response: We concur with the reviewer’s assessment and have noted this over-representation of more highly-resourced people in the limitations. We have updated the limitations section as follows (lines 481-486):

“Individuals from communities without reliable access to the internet or other low-resourced individuals may be underrepresented in this study. Finally, the data for this study were elicited from focus group discussions, which may not be representative or generalizable to broader populations. Additionally, due to small subpopulation sample sizes, in-depth exploration of themes among any one group was not possible.” 

Explain the idea of “vaccine passports” (p 13, last para)

Authors’ response: We have added the following text to elaborate on the concept of vaccine passports (lines 333-337):

“In addition to their overall resistance to prescriptive guidance, participants also feared possible further restrictions, such as community settings requiring “vaccine passports,” where access to certain venues could be dependent on providing proof of having received a COVID-19 vaccine. Vaccine passports were perceived as a basic infringement on individual liberty.”

Confirm how the data analysis led to this finding – “Liberal-leaning participants particularly expressed their willingness to help others find and obtain vaccines.” What is the evidence to support this statement? (p 15, para 1)

Authors’ response: We have clarified this statement as follows (lines 367-368):

“Among the participants who expressed their willingness to help others find and obtain vaccines, the majority identified as liberal-leaning.”

DISCUSSION

Expand the discussion of the “Increasing Vaccination Model” and show, not just tell, the connection between study results and the model. (pp 15-16)

Authors’ response: We have added the following text to illustrate how the themes correspond to the Behavioral and Social Drivers of Immunization framework (lines 387-397):

“The elicited themes primarily centered around the two domains from the BeSD Framework most associated with vaccine hesitancy, or motivational conflict thinking and feeling issues—notably, distrust of experts and institutions promoting COVID-19 vaccines and worries about the long-term effects and safety of COVID-19 vaccines, social processes—notably, resistance toward prescriptive guidance, social restrictions, and social pressure regarding vaccination, but also acceptance, and in some cases encouragement, of others getting vaccinated. These findings underscore the important role of considering intra- and inter-personal level factors on COVID-19 vaccination uptake above and beyond access, availability, and other “practical” issues.”

Define “vaccine hesitancy spectrum” (p 10, para 1) – Neither the introduction nor the discussion includes an explanation of this framework.

Authors’ response: We have moved the measure for vaccine hesitant to the methods section (lines 115-117):

“Participants who indicated that they “definitely will not” get, “probably will not” get, or were “not sure” if they would get a COVID-19 vaccine were considered vaccine hesitant. ”

In addition, we have clarified the sentence mentioned by the reviewer, as follows (lines 244-247):“Vaccine hesitant participants questioned the long-term safety of COVID-19 vaccines—particularly the Moderna and Pfizer vaccines—given their uncertainty about the novel technology used to develop them.”.”

Relate finding to the existing COVID-19 vaccine hesitancy literature – Does the study’s findings and recommendations resonate or not with other relevant research? As of now, a Pub Med search for “COVID-19 vaccine hesitancy United States” produces 462 results, and the article’s bibliography only includes ~8 other studies. The authors should embed their work much more thoroughly in the larger research. For example:

• Are there other studies that could strengthen this proposal (p 16, para 1) – “Messages about COVID-19 vaccines that provide clear, fact-based responses to the types of questions people have about topics like the vaccine approval process, the low risk of short- or long-term side effects, and the interrelationship between vaccination and NPIs such as social distancing, wearing face masks, and frequent and washing may help improve confidence in COVID-19 vaccine safety and effectiveness.”

• Do any other COVID-19 vaccine hesitancy studies have similar finding (p 16, para 2) – “[S]ome participants felt that they were already doing their part for their community by practicing social distancing and mask-wearing and thus felt that vaccination should be a personal decision rather than one based on social responsibility.”

Authors’ response: we have incorporated additional references throughout this manuscript to highlight the substantial contributions to the literature that have been published. For the specific section outlined, we have added a specific mention of time trends in risk behaviors by vaccination status from a nationally representative survey conducted April-July 2021.

Additionally, some participants felt that they were already doing their part for their community by practicing social distancing and mask-wearing and thus felt that vaccination should be a personal decision rather than one based on social responsibility, consistent with the national trends in less risk behaviors among unvaccinated adults in the US during this time (Agaku et. al. 2022).

Refrain from over-generalizing – “These findings indicate that societal pressure or messaging based on protecting one’s community may not be effective in increasing COVID-19 vaccination.” (p 16, para 2) Is this always true? What might other COVID-19 vaccine related (and other) research suggest?

Authors’ response: We have removed this statement from the manuscript. 

Resolve the apparent contradiction between the following claims (p 16, para 2-3) –

• “These findings indicate that societal pressure or messaging based on protecting one’s community may not be effective in increasing COVID-19 vaccination.”

• “…an effective approach for COVID-19 vaccination could be to encourage vaccination behavior that aligns with protecting themselves as well as their family, friends, and others in their immediate social networks.”

Authors’ response: While these ideas distinguish between pressure to comply and an empowering behavior to benefit oneself and others, we have opted to remove the first statement from the manuscript and have revised the second statement, as follow (lines 416-420)s:

“In line with findings of previous studies on HPV vaccination and physical activities [18, 19], an effective approach for COVID-19 vaccination could be to reframe vaccination from societal pressure to an empowering behavior to protect family, friends, and others in their immediate social networks [17, 20]. Clarify the logic in the sequence of your ideas –

• What is the logical step missing between these 2 sentences (p 17, para 1): “Furthermore, some participants expressed that they did not understand how vaccination was more effective than the NPI behaviors they were already practicing or why NPIs were still needed for vaccinated persons. The focus group findings suggest that information should be provided that emphasizes the essential role vaccines play in preventing severe illness and death (Griffith et al., 2010; Haldane et al., 2019). “

Authors’ response: we have added a clause to highlight the importance of communication on the complementary role of NPIs to preventing illness, as well as the essential role of COVID-19 vaccines in preventing severe illness and death (lines 425-427):

“The focus group findings suggest that information should be provided that emphasizes the complementary role of NPIs to the essential role vaccines play in preventing severe illness and death (Griffith et al., 2010; Haldane et al., 2019).”

• How does this order of ideas make sense (p 17, para 2): “In engaging non-Hispanic Black persons, recognizing past public health failures by the government and medical communities may be a positive step in building trust (Momplaisir et al., 2021; Pichon et al., 2012). Such messengers could emphasize the relatively low incidence of severe side effects identified even after hundreds of millions of COVID-19 vaccine doses have been administered in the US and globally (Gee et al., 2021).

Authors’ response: we have added a clause to clarify the importance of first acknowledging past failures, listening to concerns, and then providing evidence.

“In engaging non-Hispanic Black persons, recognizing past public health failures by the government and medical communities may be a positive step in building trust [15-17, 31, 33]. After acknowledging past failures, trusted messengers could then listen to ongoing vaccination concerns [34, 35] and engage trusted community leaders to provide responsive, tailored information to address the concerns [36].”

---

## [Editor Report · Decision Letter 1]

25 Jan 2023

A qualitative study of behavioral and social drivers of COVID-19 vaccine confidence and uptake among unvaccinated Americans in the US April-May 2021

PONE-D-22-12427R1

Dear Dr. Abad,

We’re pleased to inform you that your manuscript has been judged scientifically suitable for publication and will be formally accepted for publication once it meets all outstanding technical requirements.

Kind regards,

Lorien Shana Jasny

Academic Editor

PLOS ONE
---

## [Editor Report · Acceptance letter]

2 Feb 2023

PONE-D-22-12427R1 

A qualitative study of behavioral and social drivers of COVID-19 vaccine confidence and uptake among unvaccinated Americans in the US April-May 2021 

Dear Dr. Abad:

I'm pleased to inform you that your manuscript has been deemed suitable for publication in PLOS ONE. Congratulations! Your manuscript is now with our production department. 

Kind regards, 

on behalf of

Dr. Lorien Shana Jasny 

Academic Editor

PLOS ONE